# THE UNREASONABLE EFFECTIVENESS OF LINEAR PREDICTION AS A PERCEPTUAL METRIC

**Daniel Severo**[*]
University of Toronto
Vector Institute for A.I.
d.severo@mail.utoronto.ca

**Lucas Theis**
Google Deepmind
theis@google.com

**Johannes Ballé**
Google Research
jballe@google.com

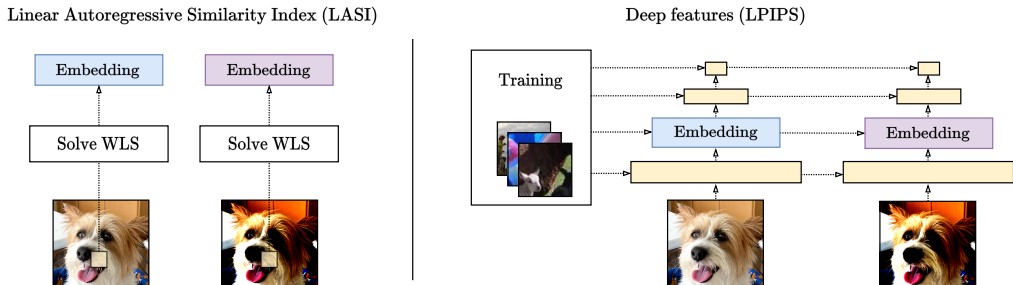

Figure 1: Comparison between our method, *Linear Autoregressive Similarity Index* (LASI), and Learned Perceptual Image Patch Similarity (LPIPS) from Zhang et al. (2018). Our method solves a weighted least squares (WLS) problem to compute perceptual embeddings, at inference time, with no prior training or neural networks. LPIPS uses pre-trained features from deep models as image embeddings and trains on human annotated data.

## ABSTRACT

We show how perceptual embeddings of the visual system can be constructed at inference-time with no training data or deep neural network features. Our perceptual embeddings are solutions to a weighted least squares (WLS) problem, defined at the pixel-level, and solved at inference-time, that can capture global and local image characteristics. The distance in embedding space is used to define a perceptual similarity metric which we call *LASI: Linear Autoregressive Similarity Index*. Experiments on full-reference image quality assessment datasets show LASI performs competitively with learned deep feature based methods like LPIPS (Zhang et al., 2018) and PIM (Bhardwaj et al., 2020), at a similar computational cost to hand-crafted methods such as MS-SSIM (Wang et al., 2003). We found that increasing the dimensionality of the embedding space consistently reduces the WLS loss while increasing performance on perceptual tasks, at the cost of increasing the computational complexity. LASI is fully differentiable, scales cubically with the number of embedding dimensions, and can be parallelized at the pixel-level. A Maximum Differentiation (MAD) competition (Wang & Simoncelli, 2008) between LASI and LPIPS shows that both methods are capable of finding failure points for the other, suggesting these metrics can be combined. Code: https://github.com/dsevero/Linear-Autoregressive-Similarity-Index.

## 1 INTRODUCTION

The applicability of computer vision in real world applications hinges on how well the loss function aligns with the human visual system. Learning end-to-end solutions for applications such as super-resolution and lossy compression (Ballé et al., 2016; 2018) requires differentiable similarity metrics

---

[*]Work done while at Google Research

that correlate well with how humans perceive change in visual stimuli. Unfortunately, widely used metrics such as PSNR/MSE that measure change at the pixel-level, although differentiable, do not satisfy this criterion.

The failure of pixel-level metrics in capturing perception has prompted the design of similarity metrics at the patch level, inspired by a subfield of human psychology known as psychophysics. The most successful one to date is the *multi-scale structural similarity metric* MS-SSIM (Wang et al., 2003; 2004) which models luminance and contrast perception. Despite these efforts, the complexity of the human visual system remains difficult to model by hand; evidenced by the failures of MS-SSIM in predicting human preferences in standardized image quality assessment (IQA) experiments (Zhang et al., 2018).

To move away from handcrafting similarity metrics the community has shifted towards using deep features from large pre-trained neural networks. For example, the *Learned Perceptual Image Patch Similarity* (LPIPS) (Zhang et al., 2018) metric assumes the L2 distance between these deep features can capture human perception. In the same work, the authors introduce the *Berkeley Adobe Perceptual Patch Similarity* (BAPPS) dataset, which has become widely accepted as a benchmark for measuring perceptual alignment of similarity metrics. LPIPS uses deep features as inputs to a smaller neural network model that is trained on the human annotated data available in BAPPS which indicate human preference of certain images over others with respect to perceived similarity. Collecting this data is expensive as it requires large-scale human trials, and the generalization capabilities of metrics beyond this dataset are not well understood (Kumar et al., 2022).

To side-step expensive data collection procedures recent work has attempted to directly learn embeddings inspired by well known phenomena of the human visual system. For example, the *Perceptual Information Metric* (PIM) (Bhardwaj et al., 2020) optimizes a mutual information (Cover, 1999) based metric and does not use annotated labels. The deep features resulting from this procedure perform competitively with LPIPS on the BAPPS dataset as well as other settings (Bhardwaj et al., 2020). Other methods such as (Wei et al., 2022) define a self-supervised objective where the neural network must predict a label indicating which distortion type, from a predefined set, was used to corrupt the image.

**In this work, we put into question the necessity of deep features to define similarity metrics aligning with human preference.** We take inspiration from recent work in the field of psychology which provide evidence that the visual working memory performs *compression* of visual stimuli (Bays et al., 2022; Bates & Jacobs, 2020b; Sims, 2018; Brady et al., 2009; Sims, 2016; Bates & Jacobs, 2020a). We employ methods that *compress a visual stimuli at inference time* with no pre-training or prior knowledge of the data distribution. Applying this procedure at inference-time means we do not require *any* expensive labelling procedures, nor unlabelled data, as in LPIPS or PIM. Our embeddings are learned at the pixel-level, but can capture patch-level semantics by solving a weighted least squares (WLS) problem from a neighborhood surrounding the pixel, a subcomponent of the lossless compression algorithm developed by Meyer & Tischer (2001).

Our *Linear Autoregressive Similarity Index* (LASI) uses the L2 norm of the differences between embeddings of images, averaged over all pixels, to define a perceptual similarity metric. We find that increasing the neighborhood size, which corresponds to the final embeddings dimensionality, consistently improves the WLS loss as well as performance on the tasks in BAPPS. This is in contrast to learned methods like LPIPS, where performance on perceptual tasks can correlate negatively with the classification performance from which the deep features are taken (Kumar et al., 2022).

An overview of full-reference image quality assessment (FR-IQA) is provided in Section 2, while Section 3 reviews a representative sample of current state-of-the-art FR-IQA algorithms. Computing the embeddings as well as LASI is discussed, and an algorithm is given, in Section 4. LASI is benchmarked against LPIPS, PIM, and A-DISTS (Zhu et al., 2022) across 6 categories of image quality experiments in Section 5. In Section 5.3, we employ the Maximum Differentiation (MAD) competition (Wang & Simoncelli, 2008) to show that LPIPS and LASI can potentially be combined, as one can be used to find failure modes of the other (see the section for a formal definition).

## 2 FULL-REFERENCE IMAGE QUALITY ASSESSMENT (FR-IQA)

FR-IQA is an umbrella term for methods and datasets designed to evaluate the quality of a distorted image, relative to the uncorrupted reference, in a way that correlates with human perception. Correlation is measured through benchmark datasets created by collecting data from psychophysical experiments such as *two-alternative forced choice* (2-AFC) human trials.

In 2-AFC image quality assessment experiments, subjects are forced to decide between two mutually exclusive alternatives relating to the perceived quality of images. For example, in Zhang et al. (2018) subjects are shown 3 images, a reference and 2 alternatives, and must indicate which of the 2 alternatives they perceive as being more similar to the reference. *Just-noticeable differences* (JND) is another type of 2-AFC experiment where two similar images are shown and subjects must generate a binary label indicating if they perceive the images as being the same or distinct.

The response of subjects in 2-AFC trials are taken to be the ground truth. For example, if 2 images in a JND dataset have different pixel values, but are judged to be the same by all subjects, then a perfect FR-IQA algorithm must also decide they are the same (Duanmu et al., 2021). In the case where there is disagreement between subjects on the same pair of images, then the uncertainty is considered inherent to human perception (i.e., *aleatoric*, not *epistemic*).

FR-IQA methods largely ignore perceptual uncertainty and instead attempt to learn a *distance function* between images that assigns small values to perceptually similar images. Algorithms can be categorized into *data-free*, *unsupervised*, and *supervised*, depending on what training data is needed to learn the distance function. Supervised methods require collecting annotated data from psychophysical experiments using human trials (e.g., LPIPS from Zhang et al. (2018)). Unsupervised methods can learn directly from unlabelled data (e.g., PIM from Bhardwaj et al. (2020)), while data-free methods require no data or training at all (e.g., MS-SSIM of Wang et al. (2003) and our method). These methods are trained and evaluated by performing train–test splits on benchmark 2-AFC datasets such as the *Berkeley Adobe Perceptual Patch Similarity* (BAPPS) (Zhang et al., 2018).

## 3 RELATED WORK

In this section we review closely related literature for data-free and learned (both unsupervised and supervised) full-reference image quality assessment (FR-IQA). For an in-depth survey see Duanmu et al. (2021); Ding et al. (2021).

Data-free distortion metrics operating at the pixel level such as mean squared error (MSE) are commonly used in lossy compression applications (Cover, 1999) but have long been known to correlate poorly with human perception (e.g., Girod, 1993). Patch-level metrics have been shown to correlate better with human judgement on psychophysical tasks. Most notably, the *Structural Similarity Index* (SSIM), as well as its multi-scale variate MS-SSIM, compare high level patch features such as luminance and contrast to define a distance between images (Wang et al., 2004). SSIM is widely used in commercial television applications, and MS-SSIM is a standard metric for assessing performance on many computer vision tasks. The method presented in this work is also data-free and outperforms MS-SSIM on benchmark datasets.

Many learned FR-IQA methods are designed mirroring the *learned perceptual image patch similarity* (LPIPS) method of Zhang et al. (2018), where a neural network is trained on some auxiliary task and the intermediate layers are taken as perceptual representations of an input image. An unsupervised distance between images is defined as the L2 norm of the difference between their representations. A supervised distance uses the representations as inputs to a second model that is trained on human annotated data regarding the perceptual quality of the input images (e.g., labels of 2-AFC datasets discussed in Section 2). Taking representations from neural networks that perform well on their auxiliary task does not guarantee good performance on perceptual tasks (Kumar et al., 2022), making it difficult to decide which existing models will yield perceptually relevant distance functions. In contrast, for the same experimental setup, the performance of our method on perceptual tasks correlated well with performance on the auxiliary task (see Section 5.1).

Self-supervision was used by Madhusudana et al. (2022b;a) and Wei et al. (2022) for unsupervised and supervised FR-IQA. Images are corrupted with pre-defined distortion functions and a neural network is trained with a contrastive pairwise loss to predict the distortion type and degree. The

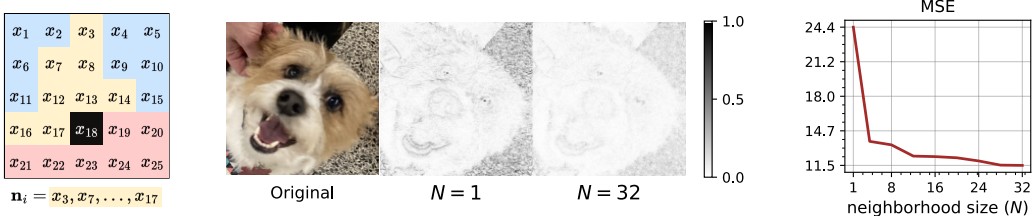

Figure 2: **Left:** Definition of previous pixels $\mathbf{x}_{[1,i)} = (x_1, \ldots, x_{17})$, obeying raster-scan ordering, and causal neighborhood $\mathbf{n}_i$ (in yellow) of pixel $x_i$ for $i = 18$. The image has dimensions $5 \times 5 \times 1$. $\mathbf{n}_i$ is defined as the $N = 8$ closest pixels to $x_{18}$ in terms of L1 distance. For example, the distance from pixel $x_1$ to $x_{18}$ is 5, while from $x_{12}$ to $x_{18}$ it is 2. Pixels in the red region are not considered as they come after the 18-th pixel in the ordering. As the neighborhood size $N$ increases, pixels with smaller L1 distance are added to $\mathbf{n}_i$ in the same order they appear in $\mathbf{x}_{[1,i)}$. **Middle:** An image of dimension $256 \times 256 \times 3$ and the squared residuals of the prediction with Equation (1). **Right:** MSE of images reconstructed with Equation (1).

unsupervised distance is defined as discussed previously and ridge regression is used to learn a supervised distance function. This method requires training data, while our method requires no training at all.

## 4 METHOD

Here we present our data-free, self-supervised (at inference time) FR-IQA algorithm called *Linear Autoregressive Similarity Index* (LASI). We make use of a sub-component of the lossless compression algorithm available in Meyer & Tischer (2001) to define a distance between images, which is described in detail next.

### 4.1 CONSTRUCTING PERCEPTUAL EMBEDDINGS VIA WEIGHTED LEAST SQUARES

Our method relies on self-supervision (at inference-time) to learn a representation for each pixel that captures global perceptual semantics of the image. The underlying assumption is that, given a representation vector for some pixel, it must successfully predict the value of *other pixels* in the image in order to capture useful semantics of the image's structure. Our method acts directly on images $\mathbf{x}, \mathbf{y}$ to compute a distance $d(\mathbf{x}, \mathbf{y})$, similar to other data-free methods like L2 and MS-SSIM (Wang et al., 2003). We describe this formally next.

**Weighted Least Squares** Let $\mathbf{x} = (x_1, \ldots, x_k) \in \mathbb{R}^k$ represent a flattened image with height $H$, width $W$, number of channels $C$, and $k = HWC$ pixels. Our method is autoregressive and uses the previous $i - 1$ pixels $\mathbf{x}_{[1,i)} = (x_1, \ldots, x_{i-1})$ to predict the value $x_i$ of the $i$-th pixel. The number of previous pixels used will be equal to the dimensionality of the embeddings. Therefore, we restrict the algorithm to use a subset $N \le i - 1$ of pixels from $\mathbf{x}_{[1,i)}$. The subset is made up of the elements in $\mathbf{x}_{[1,i)}$ that are closest [1] in the coordinate space of the image to the $i$-th pixel. We refer to this as the *causal neighborhood* of pixel $i$, and represent it as a vector $\mathbf{n}_i \in \mathbb{R}^N$. See Figure 2 for an example.

For the $i$-th pixel of value $x_i$, we find a vector $\mathbf{w}_i(\mathbf{x}_{[1,i)}) \in \mathbb{R}^N$ that minimizes the weighted least squares objective

$$\mathbf{w}_i(\mathbf{x}_{[1,i)}) = \arg\min_{\mathbf{w} \in \mathbb{R}^N} \sum_{j < i} \omega^{\ell_{ij}} \left( \mathbf{n}_j^\top \mathbf{w} - x_j \right)^2, \tag{1}$$

where $0 < \omega \le 1$ is a hyperparameter and $\ell_{ij}$ the Manhattan distance between coordinates of the $i$-th and $j$-th pixels in the image (see Figure 2 for an example). Concatenating $\mathbf{w}_i(\mathbf{x}_{[1,i)})$ column-wise yields the perceptual embedding matrix $\mathbf{W}(\mathbf{x}) \in \mathbb{R}^{N \times k}$ of image $\mathbf{x}$.

---

[1]In Manhattan distance.

Equation (1) defines a (weighted) least squares problem with data points $\{(\mathbf{n}_j, x_j)\}_{j=1}^{i-1}$ extracted from the previous pixels. The weights $\omega^{\ell_{ij}}$ decrease as the distance $\ell_{ij}$ between coordinates increases, biasing the objective to better predict closer pixels. The value of $x_i$ is *not* used to compute the representation $\mathbf{w}_i(\mathbf{x}_{[1,i)})$ but is used in the computation of subsequent representations.

**Distance Function**    The LASI distance between images is defined as the distance between their perceptual embeddings averaged over pixels $d(\mathbf{x}, \mathbf{y}) = \frac{1}{k} \sum_{i=1}^{k} \|\mathbf{w}_i(\mathbf{x}_{[1,i)}) - \mathbf{w}_i(\mathbf{y}_{[1,i)})\|_2$.

**Differentiability**    All operations, including solving Equation (1), are differentiable which allows us to compute the gradients of $d$ with respect to both arguments. In Section 5.3 we use differentiability to perform the Maximum Differentiation (MAD) Competition (Wang & Simoncelli, 2008) between our method and LPIPS (Zhang et al., 2018).

**Predictions**    Meyer & Tischer (2001) solve Equation (1) to generate a prediction $\hat{x}_i = \mathbf{w}_i(\mathbf{x}_{[1,i)})^\top \mathbf{n}_i$ of the $i$-th pixel which is then used for lossless compression of the original image. Figure 2 shows examples of the squared residual image made up of pixels $z_i = (x_i - \hat{x}_i)^2$ for varying sizes of neighborhood size $N$. In Section 5.1, Figure 3, we show the prediction loss $\sum_{i=1}^{k} z_i^2$ has strong correlation with performance on downstream 2-AFC tasks.

## 4.2    ALGORITHM

Here we describe our implementation which solves (1) in 3 steps. The algorithm is differentiable and most operations can be run in parallel on a GPU. Compute time and memory can be traded-off by, for example, precomputing the rank-one matrices. It is also possible to solve Equation (1) thrice in parallel, once for each channel, and average the results at the expense of some performance on downstream perceptual tasks.

The steps of our method are:

**1) Transform**    For the $i$-th pixel of value $x_i$, compute a rank-one matrix from the neighborhood, as well as another vector equal to the neighborhood scaled by the pixel itself:

$$\mathbf{A}_i = \mathbf{n}_i \mathbf{n}_i^\top \in \mathbb{R}^{N \times N}, \qquad \mathbf{b}_i = x_i \mathbf{n}_i \in \mathbb{R}^N. \tag{2}$$

**2) Weigh-and-Sum**    On a second pass, for each pixel, compute a weighted sum of the rank-one matrices of all previous pixels, weighted by $\omega^{\ell_{ij}}$, where $0 < \omega < 1$ is a hyperparameter and $\ell_{ij}$ the Manhattan distance between locations of pixels $x_i$ and $x_j$. Perform a similar procedure for vectors $\mathbf{b}_i$:

$$\bar{\mathbf{A}}_i = \sum_{j=1}^{i-1} \omega^{\ell_{ij}} \mathbf{A}_j, \qquad \bar{\mathbf{b}}_i = \sum_{j=1}^{i-1} \omega^{\ell_{ij}} \mathbf{b}_j. \tag{3}$$

**3) Solve**    Finally, for each pixel, solve the least-squares problem with coefficients $\bar{\mathbf{A}}_i$ and target vector $\bar{\mathbf{b}}_i$ by computing the Moore-Penrose pseudo-inverse $\bar{\mathbf{A}}_i^\dagger$ of $\bar{\mathbf{A}}_i$,

$$\mathbf{w}_i(\mathbf{x}_{[1,i)}) = \bar{\mathbf{A}}_i^\dagger \bar{\mathbf{b}}_i. \tag{4}$$

The rank-one matrices $\bar{\mathbf{A}}_i$ have dimension $N \times N$. In our experiments, we found $N = 12$ was sufficient to perform competitively with unsupervised methods on images of size $64 \times 64 \times 3$. In this regime of small $N$ computing the pseudo-inverse can be done directly using the singular value decomposition (SVD) (Klema & Laub, 1980) (we use `jax.numpy.linalg.pinv`; Bradbury et al., 2018).

**Computational Complexity**    Solving Equation (4) requires computing the SVD of $\bar{\mathbf{A}}_i$ which has worst case complexity of $\mathcal{O}(N^3)$ (Klema & Laub, 1980), where $N$ is the neighborhood size and embedding dimensionality. This must be done for each pixel but can be parallelized at the expense of an increase in memory, with no loss in performance. Figure 3 shows LASI is faster than PIM, which requires forward-passes in a neural network, on the BAPPS dataset.

|  | | BAPPS-2AFC | | | | | | BAPPS-JND | |
|---|---|---|---|---|---|---|---|---|---|
|  | Metric | Trad. | CNN | SRes. | Deblur. | Color. | Interp. | Trad. | CNN |
| Ref. | Majority | 85.8 | 88.5 | 80.1 | 74.0 | 76.4 | 76.0 | — | — |
|  | Human | 80.8 | 84.3 | 73.6 | 66.1 | 68.8 | 68.6 | — | — |
| Supervised | LPIPS | 79.3 | 83.7 | 71.5 | 61.2 | 65.6 | 63.3 | 51.9 | 67.9 |
| Unsuperv. | LPIPS (fts. only) | 73.3 | **83.1** | **71.7** | 60.7 | **65.0** | **62.7** | 46.9 | **67.9** |
|  | A-DISTS | — | — | 70.8 | 60.2 | 62.1 | 61.6 | — | — |
|  | PIM | **75.7** | 82.7 | 70.3 | **61.6** | 63.3 | 62.6 | **57.8** | 60.1 |
| Data-free | L2 | 59.9 | 77.8 | 64.7 | 58.2 | 63.5 | 55.0 | 33.7 | 58.2 |
|  | MS-SSIM | 61.2 | 79.3 | 64.6 | 58.9 | 57.3 | 57.3 | 36.2 | **63.8** |
|  | LASI (Ours) | **73.8** | **81.4** | **70.2** | **60.5** | 62.1 | 62.3 | 55.9 | 63.2 |
| % Diff. to best Unsuperv. | | -2.5 | -2.0 | -2.1 | -1.8 | -4.5 | -0.6 | -3.3 | -5.8 |
| % Diff. to Supervised LPIPS | | -6.9 | -2.7 | -1.9 | -1.2 | -5.3 | -1.6 | +9.3 | -7.4 |
| % Diff. to MS-SSIM | | +20.6 | +2.6 | +8.7 | +2.7 | +8.4 | +8.7 | +54.4 | -0.9 |

Table 1: Results for just-noticeable differences (BAPPS-JND) and two-alternative forced choice (BAPPS-2AFC) experiments (higher is better) of the BAPPS dataset (Zhang et al., 2018). Numbers in gray are reference values. The first two entries under Ref. are theoretically calculated references. Majority is the highest score that can be achieved in each column while Human represents the average agreement between two randomly selected subjects. LPIPS (supervised) is trained on human annotated labels. Unsupervised methods require unlabelled examples for training, while data-free methods do not require training data at all. "% Diff. to best Unsuperv." is the percentage difference between our method and the cherry-picked best performing unsupervised method shown in bold. Supervised LPIPS is trained on the same distortion types present in "Trad." and "CNN" and therefore has an unfair advantage relative to other metrics in these categories (Zhang et al., 2018). These values are indicated in gray in the row labeled "% Diff. to Supervised LPIPS". See Section 2 and Section 5 for more details.

## 5 EXPERIMENTS

In this section we compare our method against state-of-the-art unsupervised FR-IQA algorithms. Our method is data-free but performs competitively with learned methods on experiments of the *Berkeley Adobe Perceptual Patch Similarity* (BAPPS) dataset (Zhang et al., 2018) (the first row of Figure 4 shows examples from BAPPS). Experiments of Figure 3 indicate the prediction loss and performance on perceptual tasks correlate and improve as the neighborhood size increases.

In Table 1, results for PIM as well as LPIPS numbers for BAPPS-JND are taken from Table 1 of Bhardwaj et al. (2020). For LPIPS, we generously took the best models from Table 5 of Zhang et al. (2018) for each 2-AFC category. For our method we used $N = 12$ across all experiments and the decay parameter in Equation (1) was fixed to $\omega = 0.8$, the same value used for the lossless compression algorithm of Meyer & Tischer (2001). Images in all categories have dimensions $64 \times 64 \times 3$ which is 1536 times larger than the neighborhood size.

The performance consistently improves with larger neighborhood sizes $N$ as can be seen from the solid line in Figure 3. This suggests the parameter can be used to trade off computational complexity and performance and does not require tuning.

The last row of Table 1 shows the gap in performance between our data-free method and the best performing unsupervised neural network approach. **In the worst case, our method scores only** $5.8\%$ **less while requiring no learning, data collection, or expensive neural network models.**

### 5.1 TWO-ALTERNATIVE FORCED CHOICE (BAPPS-2AFC)

This section discusses experiments on the BAPPS-2AFC dataset of Zhang et al. (2018). BAPPS-2AFC was constructed by collecting data from a 2-AFC psychophysical experiment where subjects are shown reference images $\mathbf{r}^{(\ell)}$ as well as alternatives $\mathbf{x}_0^{(\ell)}, \mathbf{x}_1^{(\ell)}$ and must decide which of the two alternatives they perceive as more similar to the reference. The share of subjects that select image $\mathbf{x}_1^{(\ell)}$ over $\mathbf{x}_0^{(\ell)}$ is available in the dataset as $p^{(\ell)}$.

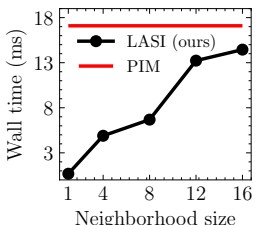 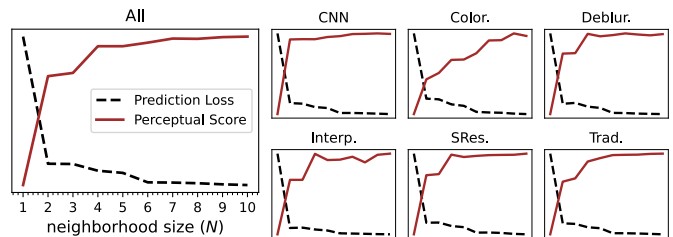

Figure 3: **Left:** Total time to compute distances for 1000 examples from BAPPS on an NVIDIA V100. Results are averaged over 100 runs. **Right:** Correlation of performances on the self-supervised prediction task and perceptual 2-AFC task. Both curves are normalized to lie between 0 and 1 for exposition. The prediction loss (dashed line) is calculated by computing the MSE between the original and reconstructed reference image. As the causal neighborhood size $N$ increases, the prediction loss decreases (lower is better) as the performance on perceptual tasks increases (higher is better). The curves on the left plot are computed using all examples in the BAPPS-2AFC dataset (Zhang et al., 2018), while the right plots are broken down by sub-categories of the same dataset.

The performance of a FR-IQA algorithm, defined by a distance function $d$ between images, on a 2-AFC dataset is measured by

$$\frac{1}{n}\sum_{\ell=1}^{n}\left(p^{(\ell)}\right)^{a^{(\ell)}}\left(1-p^{(\ell)}\right)^{1-a^{(\ell)}} \leq \frac{1}{n}\sum_{\ell=1}^{n}\max\{p^{(\ell)}, 1-p^{(\ell)}\}, \quad (5)$$

where $a^{(\ell)} = \mathbf{1}\left[d(\mathbf{x}_1^{(\ell)}, \mathbf{r}^{(\ell)}) < d(\mathbf{x}_0^{(\ell)}, \mathbf{r}^{(\ell)})\right]$ [2]. Equality is achieved when the algorithm agrees with the majority of subjects, i.e. $a^{(\ell)} = \lfloor p^{(\ell)} \rceil$ [3], for all examples in the dataset. Human-level performance is defined as

$$\frac{1}{n}\sum_{\ell=1}^{n}\left[\left(p^{(\ell)}\right)^2 + \left(1-p^{(\ell)}\right)^2\right], \quad (6)$$

which corresponds to the average probability of two randomly chosen subjects agreeing. Majority and human-level performance scores for our 2-AFC experiments are shown in the first rows of Table 1.

Across all categories our method scored competitively with the best performing unsupervised method, with the largest gap being $4.5\%$ in the "Colorization" category. Our method outperforms MS-SSIM (Wang et al., 2003) on all 2-AFC categories, most notably in "Traditional" where the improvement is $20.6\%$, and provides perceptual embeddings (Equation 1) for use in downstream computer vision tasks. We highlight the overall improvements with respect to MS-SSIM in the last row of Table 1.

**Correlation with Self-Supervised Task** In an empirical study, Kumar et al. (2022) investigate if deep features from better performing classifiers achieve better perceptual scores on the BAPPS dataset. Surprisingly, their results suggest the correlation can be negative: more accurate models produce embeddings that capture less meaningful perceptual semantics, performing poorly on 2-AFC and JND tasks when used together with LPIPS. We perform a similar study with our method on the prediction task defined in Meyer & Tischer (2001) as a function of neighborhood size. For each reference image $\mathbf{r}^{(\ell)}$, we compute the prediction $\hat{\mathbf{r}}^{(\ell)}$, composed of pixels $\hat{r}_i^{(\ell)} = \mathbf{w}_i(\mathbf{r}_{[1,i)})^{\top}\mathbf{n}_i$, and report the residual $\frac{1}{n}\sum_{\ell=1}^{n}(r_i^{(\ell)} - \hat{r}_i^{(\ell)})^2$ alongside the score on the 2-AFC task. Results are shown in Figure 3. The performance on the prediction and perceptual tasks show a strong correlation, and improve consistently, across all 2-AFC categories.

## 5.2 JUST-NOTICEABLE DIFFERENCES (BAPPS-JND)

In Table 1 we compare our data-free method to recent unsupervised methods on the JND subset of the BAPPS dataset. The BAPPS-JND dataset $\{(\mathbf{x}^{(i)}, \tilde{\mathbf{x}}^{(i)}, p^{(i)})\}_{i=1}^{n}$ was created by showing two images

---

[2] $\mathbf{1}[A] = 1$ iff statement $A$ is true.
[3] $\lfloor \cdot \rceil$ rounds to the nearest integer

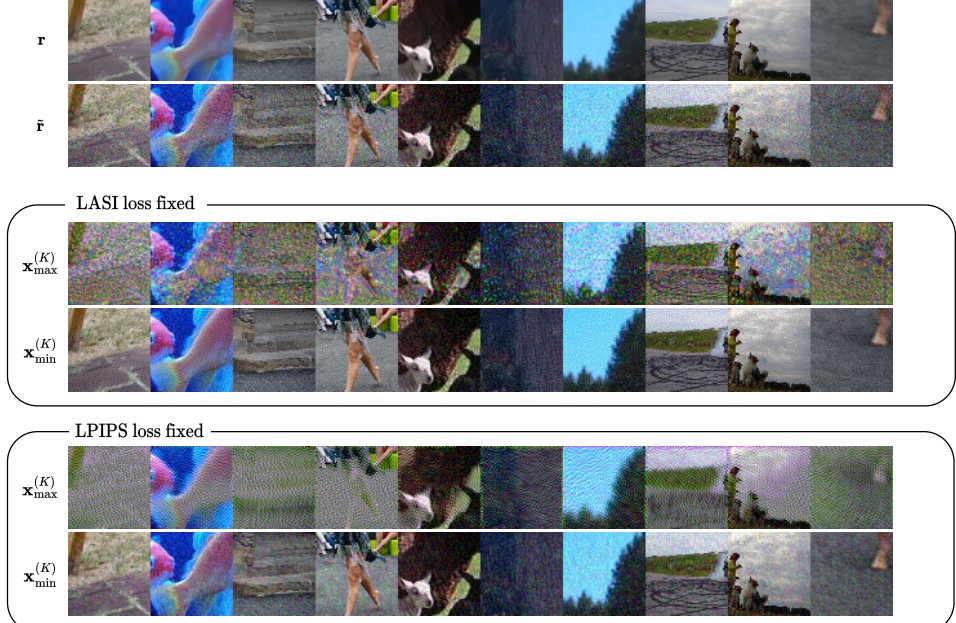

Figure 4: Results from the MAD competition (Wang & Simoncelli, 2008) between LASI (ours) and LPIPS (Zhang et al., 2018). The first 10 images from the "Interp." category of BAPPS-JND are shown in the first row labelled $\mathbf{r}$. Gaussian noise was added to $\mathbf{r}$ to create images $\tilde{\mathbf{r}}$ shown in the second row. Images in the same column of the middle rows in "LASI loss fixed" are equidistant to their corresponding references $\mathbf{r}$, as measured by LASI. The same is true for the bottom rows under LPIPS. These examples constitute failure points of LASI and LPIPS, as images in the bottom rows of each box ($\mathbf{x}_{\min}^{(K)}$) are perceptually closer to their references $\mathbf{r}$, yet have the same distance to $\mathbf{r}$ as the distorted images in the top rows ($\mathbf{x}_{\max}^{(K)}$).

$\mathbf{x}^{(i)}, \tilde{\mathbf{x}}^{(i)}$ to a group of subjects where the latter is the former but corrupted by one of two different distortion types (identified by the last 2 columns of Table 1). Subjects must indicate if they perceive the images as being the same or not. The share of subjects that judged the images as being the same, $p^{(i)}$, is available in the dataset but not the individual responses. See (Zhang et al., 2018) for more details regarding the images as well as distortion types.

As described in Section 2, an FR-IQA algorithm in the context of a JND task will attempt to output a binary response that correlates with $p^{(i)}$. This defines a binary classification task where the distance defined by the FR-IQA algorithm must be thresholded to yield a decision and precision/recall can be traded-off by varying the threshold value (Bishop & Nasrabadi, 2006). We evaluate the JND experiment with an area-under-the-curve score known as mean average precision (mAP), following Zhang et al. (2018); Bhardwaj et al. (2020).

Results are shown in the last 2 columns of Table 1. For CNN-based distortions our method scores similarly to MS-SSIM (63.8% vs 63.2%). PIM loses to our method (60.1% vs 63.2%) while LPIPS outperforms it only by 5.8%.

Similar to the 2-AFC experiments, the subcategory with the largest gap between data-free and unsupervised methods is "Traditional". The gap is drastically reduced by our method by raising the highest score from 36.2% (achieved by MS-SSIM) to 55.9%, which is within 3.3% of the best scoring unsupervised method (PIM). In this same subcategory our method significantly outperforms LPIPS (55.9% vs 46.9%).

## 5.3 MAXIMUM DIFFERENTIATION (MAD) COMPETITION

MAD competition (Wang & Simoncelli, 2008) is a technique to discover failure modes in the perceptual space of differentiable similarity metrics. Failure modes are image pairs $(\mathbf{x}_1, \mathbf{x}_2)$ where

one image is clearly closer to a reference $\mathbf{r}$ upon inspection, but the metric assigns similar distances $d(\mathbf{r}, \mathbf{x}_1) \approx d(\mathbf{r}, \mathbf{x}_2)$.

We now describe the MAD competition outlined in Wang & Simoncelli (2008) that uses $K$ steps of projected gradient descent to find a failure mode $(\mathbf{x}_{\max}^{(K)}, \mathbf{x}_{\min}^{(K)})$ in $d$. First, the reference $\mathbf{r}$ is corrupted with noise yielding $\tilde{\mathbf{r}}$ which acts as the starting point $(\mathbf{x}_{\max}^{(0)}, \mathbf{x}_{\min}^{(0)}) = (\tilde{\mathbf{r}}, \tilde{\mathbf{r}})$ for optimization. At the $i$-th step, the image $\mathbf{x}_{\max}^{(i)}$ is updated using a gradient step in the direction that maximizes it's L2 distance to the uncorrupted reference $\mathbf{r}$. However, before updating, the gradient is projected into the space orthogonal to $\nabla_{\mathbf{x}_{\max}^{(i)}} d(\mathbf{r}, \mathbf{x}_{\max}^{(i)})$. This projection step guarantees that the distance to the reference does not change significantly, i.e., $d(\mathbf{r}, \mathbf{x}_{\max}^{(i)}) \approx d(\mathbf{r}, \mathbf{x}_{\max}^{(i+1)})$. The same procedure is done for $\mathbf{x}_{\min}^{(K)}$, but the gradient step is taken in the opposite direction (minimizing). In practice an extra correction step is required as the projected gradient will be tangent to the set of equidistant images (i.e., the level set). It is common to replace L2 distance with another similarity metric as a way to contrast failure modes and possibly discover ways to combine models (Wang & Simoncelli, 2008).

We performed the MAD competition between LPIPS and LASI. Qualitative results are shown in Figure 4. The neighborhood size of LASI was held fixed at $N = 16$ while deep features from VGG (Simonyan & Zisserman, 2014) were used for LPIPS as in Zhang et al. (2018). Images are parameterized by an unconstrained tensor which is then passed through a sigmoid function to yield an image in the RGB space. Gaussian noise was added in the parameter space (i.e., before the sigmoid is applied) to generate the corrupted reference $\tilde{\mathbf{r}}$, to guarantee images are valid RGB images.

Results indicate LASI can find failure points for LPIPS and vice-versa. Each metric fails in different ways. Image $\mathbf{x}_{\max}^{(K)}$ shows artifacts resembling the structure of the causal neighborhood for LASI while artifacts for LPIPS are smoother.

The difference in failure modes suggests LASI and LPIPS can be combined. Merging these models is non-trivial as the embedding dimensions differ in size making it difficult to perform simple aggregation techniques at the embedding level. We leave this as future work.

## 6 CONCLUSION

In this work we show how perceptual embeddings can be constructed at inference time with no training data or deep neural network features. Our Linear Autoregressive Similarity Index (LASI) metric performs competitively with learned methods such as LPIPS (Zhang et al., 2018) and PIM (Bhardwaj et al., 2020) on benchmark psychophysical datasets, and outperforms other untrained methods like MS-SSIM (Wang et al., 2003).

LASI solves a weighted least squares problem at inference time to create embeddings that capture meaningful semantics of the human visual system. Evidence shows increasing the embedding dimensionality improves the overall downstream performance on the tasks present in the BAPPS dataset (Zhang et al., 2018), while improving the WLS loss. This is in strong contrast to learned methods like LPIPS, where the classification performance of deep networks can correlate negatively with perception (Kumar et al., 2022).

There are many candidate hypotheses for the unreasonable effectiveness of LASI in FR-IQA, of which we discuss two. First, it is unclear how the performance of an algorithm on BAPPS generalizes to other tasks and datasets; warranting a discussion if BAPPS is indeed a valid benchmark for FR-IQA beyond small image patches. Alternatively, it is possible the performance of LPIPS and LASI are due to different reasons. While LPIPS embeddings are constructed by indirectly compressing data samples during training, LASI embeddings are tasked with compressing a specific image.

We conclude with a myriad of open directions to explore. One such direction is to investigate if LASI embeddings, i.e., the solutions to the WLS problem, have useful semantics in computer vision beyond perceptual tasks. The choice of using WLS was inspired by Meyer & Tischer (2001) who use it to perform lossless compression of grayscale images, but there are other small-scale regression tasks that could be used.

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
