# OpenReview forum: "The Unreasonable Effectiveness of Linear Prediction as a Perceptual Metric"
_ICLR.cc/2024/Conference — ICLR 2024 poster_

### Official Review · Reviewer_QSZk · 2023-10-31

**Soundness:** 3 good
**Presentation:** 3 good
**Contribution:** 3 good
**Rating:** 6
**Confidence:** 4

**Summary:**

This work focuses on solving the weighted least squares problem for image quality distance in full-reference image quality assessment. Inspired by lossless compression, this work proposes the Autoregressive Similarity Index (LASI), which obtains perceptual embeddings by calculating a weighted sum of the causal neighborhood subset of pixel values to predict the current pixel value. The performance of the data-free LASI is comparable to that of supervised methods such as LPIPS and unsupervised methods like PIM.

**Strengths:**

1.	The LASI metric is designed in a simple yet solid manner. Full-reference Image Quality Assessment (FR IQA) can be approached from the perspective of lossless compression and the semantic information within the neighborhood of pixels.
2.	The experiments of JND，2AFC and MAD are very detailed and the explanations are very clear.

**Weaknesses:**

1. The experimental dataset consists only BAPPS.
2. An ablation study is missing for the causal neighborhood and non-causal neighborhood, as semantic understanding relies on contextual relationships around pixels or regions.

**Questions:**

1. The sentence in the section 4.1 ‘’Our method relies on self-supervision (at inference-time) to learn a representation for each pixel that captures global perceptual semantics of the image. The underlying assumption is that, given a representation vector for some pixel, it must successfully predict the value of other pixels in the image in order to capture useful semantics of the image’s structure.” But the relationship of the perceptual embedding in FR-IQA and the semantic extraction is confused in this work. Because many previous methods also use semantic features extracted by pre-trained models on high-level classification tasks to calculate perceptual distance. Is there any difference between this semantic feature and the embedding derived by Eq 1 in this work?
2. Since the LASI is designed on the semantic extraction in the images’ structure, so the correlation of the prediction task and perceptual task is good, which is a little obvious.
4. Perhaps LASI can only measure perceptual distance at patch level, and will it be useful for high-resolution images?
5. An ablation study is missing for the causal neighborhood and non-causal neighborhood, as semantic understanding relies on contextual relationships around pixels or regions.

---

> ### Author Response · Authors · 2023-11-18
>
> We thank the reviewer for the questions regarding ablations and interest in the correlation between perceptual and downstream tasks, of which answers will be incorporated into the main text.
>
> We hope our rebuttal addresses your concerns and you will consider increasing your score.
>
> ---
>
> > The experimental dataset consists only BAPPS.
>
> We are aware of the existence of other full-reference (i.e. pairwise) IQA datasets, however BAPPS is the only one we know of that contains a rich set of distortions. Other established datasets such as TID2013 or LIVE do not contain CNN-based distortions such as warping, and as such are prone to overfitting to (which we believe has already happened with traditional metrics such as MS-SSIM). A topic for future work would be to collect our own data to validate on, but this is unfortunately outside of the scope of this paper.
>
> ---
>
> > Perhaps LASI can only measure perceptual distance at patch level, and will it be useful for high-resolution images?
>
> Distortions are usually applied to the image as a whole and therefore different patches are distorted in a similar way.
> It is common in the FR-IQA literature to scale up to larger images by computing the metric on smaller patches for this reason.
> LASI can be run in the same way by parallelizing over patches and averaging the distance, a common technique in the literature.
> We expect this to perform similarly, the same way other methods do, and will include an experiment on LIVE IQA [1] in the follow-up.
>
> [1] Sheikh H. LIVE image quality assessment database release 2. http://live.ece.utexas.edu/research/quality. 2005.
>
> ---
>
> > An ablation study is missing for the causal neighborhood and non-causal neighborhood, as semantic understanding relies on contextual relationships around pixels or regions.
>
> We thank the reviewer for this suggestion of what is effectively a new version of our LASI metric.
> This exact ablation has been run and is shown below.
> The performance of the new suggestion is slightly worse, however, the implementation is simplified as computing the non-causal mask consumes less memory.
>
> [Please click for ablation plot](https://postimg.cc/V5ZPm8jk)
>
> ---
>
> > The sentence in the section 4.1 ‘’Our method relies on self-supervision (at inference-time) to learn a representation for each pixel that captures global perceptual semantics of the image. The underlying assumption is that, given a representation vector for some pixel, it must successfully predict the value of other pixels in the image in order to capture useful semantics of the image’s structure.” But the relationship of the perceptual embedding in FR-IQA and the semantic extraction is confused in this work. Because many previous methods also use semantic features extracted by pre-trained models on high-level classification tasks to calculate perceptual distance. Is there any difference between this semantic feature and the embedding derived by Eq 1 in this work?
>
> We are not sure if we understand your question correctly, but we do not think that we are confusing two different concepts in this work. Computing a perceptual embedding of an image in many prior works is done via a function $f(x; \theta)$, where $x$ is the image and $\theta$ are, for example, neural network parameters that are determined by a separate training stage.
> In LASI, computing the weights is done via a function $f(x)$ that does not have trainable parameters, hence no training stage is necessary. Computing the embedding is performed via solving a linear least squares problem, and hence there is no analytic formula for $f$.
> However, that does not matter, since $f$ is still a deterministic function of the image: every time we solve for $w$ (the weights of the linear least squares problem), we get the same answer.
>
> ---
>
> > Since the LASI is designed on the semantic extraction in the images’ structure, so the correlation of the prediction task and perceptual task is good, which is a little obvious.
>
> There is no evidence in the psychophysical literature that points towards the human visual system performing low-level reconstruction tasks.
> It is not immediately obvious to us that good perceptual embeddings can be learned from linear prediction of pixels.
>
> However, we agree that it is intuitive to expect the correlation to be better for LASI (prediction) than for LPIPS (classification).
> The reason being that a useful feature for classification is not necessarily useful for visual tasks regarding the image’s structure.

---

> > ### Comment · Reviewer_QSZk · 2023-11-23
> >
> > thanks for the clarification which partly address my concerns, I will keep the score.

---

### Official Review · Reviewer_8dYU · 2023-11-01

**Soundness:** 3 good
**Presentation:** 3 good
**Contribution:** 3 good
**Rating:** 6
**Confidence:** 3

**Summary:**

The paper introduces a new perceptual metric that requires no training data nor DNN features. In particular, taking inspiration from psychology finding that visual working memory compresses visual stimuli, the proposed method, named Linear Autoregressive Similarity Index (LASI), compresses a visual stimuli during inference.

**Strengths:**

- The paper introduces a new perceptual metric that requires no training data or DNN features.

- The paper is clearly written and easy to follow.

**Weaknesses:**

- The paper claims to have competitive performance, compared with LPIPS method. But, 11% difference in SR in Table 1 seems rather high.

- The paper claims that the advantage of the proposed method is that it requires no training data or DNN features. But, the method requires computations at inference time. Considering that once training is done, DNN-feature based methods do not require much of extra computations (hence the cost is amortized) while the proposed method requires extra computations, is requiring training data or DNN features really a bad thing?

- All experiments are performed with 64x64 resolution, which seem rather small. Is the proposed method effective for larger images, compared to other works? Is the choice of $N$ the size of neighborhood robust against the image size? It seems $N$ needs to be tuned for each image resolution, which can be critical, since images can come in at various sizes.

- The paper claims that the method can be combined with LPIPS but I cannot find the experimental results on this. Does the combination actually bring improvements?

**Questions:**

Written in the weaknesses section.

---

> ### Author Response · Authors · 2023-11-18
>
> We would like to thank the reviewer for their questions regarding compute time which we will incorporate the response into the main text. LASI (ours) requires less compute than DNN methods due to their expensive forward-pass.
>
> We hope our rebuttal addresses your concerns and you will consider increasing your score.
>
> ---
>
> > The paper claims that the advantage of the proposed method is that it requires no training data or DNN features. But, the method requires computations at inference time. Considering that once training is done, DNN-feature based methods do not require much of extra computations (hence the cost is amortized) while the proposed method requires extra computations, is requiring training data or DNN features really a bad thing?
>
> Even with amortization, computing DNN features requires more computation at inference time than LASI (ours).
> DNN features require a forward-pass in the neural network to compute embeddings.
> The plot below shows the wall-time for computing PIM (a DNN) and LASI (ours) distances, compiled with `tf.function` and `jax.jit`, respectively.
> Note how LASI is faster for the neighborhood sizes considered in the experiments (i.e., $10$ to $12$).
>
> [Please click for LASI and PIM comparison plot](https://postimg.cc/ygnnh7Vg)
>
> ---
>
> > The paper claims to have competitive performance, compared with LPIPS method. But, 11% difference in SR in Table 1 seems rather high.
>
> The difference mentioned by the reviewer is with respect to supervised LPIPS, i.e., after fine-tuning LPIPS on human annotated data.
> When comparing to unsupervised LPIPS (i.e., neural network embeddings before fine-tuning), which we hold to be the fairer comparison, the difference is less than $2\\%$.
>
> We will call attention to this fact in the caption of Table 1.
>
> ---
>
> > All experiments are performed with 64x64 resolution, which seem rather small. Is the proposed method effective for larger images, compared to other works?
>
> Distortions are usually applied to the image as a whole and therefore different patches are distorted in a similar way.
> It is common in the FR-IQA literature to scale up to larger images by computing the metric on smaller patches for this reason.
> LASI can be run in the same way by parallelizing over patches and averaging the distance, a common technique in the literature.
> We expect this to perform similarly, the same way other methods do, and will include an experiment on LIVE IQA [1] in the follow-up.
>
> [1] Sheikh H. LIVE image quality assessment database release 2. http://live.ece.utexas.edu/research/quality. 2005.
>
> ---
>
> > Is the choice of $N$ the size of neighborhood robust against the image size? It seems $N$ needs to be tuned for each image resolution, which can be critical, since images can come in at various sizes.
>
> $N$ does not need to be tuned for each image resolution.
> Our experiments show increasing $N$ gives better downstream performance (see Figure 3).
> $N$ should be set to as large as possible while respecting the compute and memory budget of the application.
>
> ---
>
> > The paper claims that the method can be combined with LPIPS but I cannot find the experimental results on this. Does the combination actually bring improvements?
>
> The MAD experiment highlights that LPIPS and LASI fail in different ways.
> We experimented with merging these models by taking geometric averages of their distances.
> This resulted in a slightly better perceptual performance but not enough to justify merging, as the compute required is now that of LPIPS + LASI.
>
> Merging these models is a non-trivial manner as the embedding dimensions differ in size making it difficult to perform simple aggregation techniques at the embedding level. Note this is true in general for any pair of models.
>
> We will update the writing of the paper to make this clear in the main text.

---

> > ### Comment · Reviewer_8dYU · 2023-11-21
> >
> > I appreciate the authors' response.
> > In regards to performance comparison regarding super resolution, I do not completely agree with the authors' arguments. The paper seems to point the proposed method as an alternative of LPIPS. Regardless of finetuning or not, if fine-tuned LPIPS is commonly used, shouldn't the comparison be done against fine-tuned LPIPS. There is no difference in inference computational complexity, regardless of whether the fine-tuning is done or not. And also, why do authors think that there is such difference observed only in super-resolution task?
> >
> > Also, the paper can be revised during the discussion phase, and I suggest sharing the revision to show the reviewers how the paper would be updated.

---

> > > ### Author Response · Authors · 2023-11-21
> > >
> > > Thank you for your response.
> > >
> > > After revisiting Table 5 on page 12 of [1]  we noticed the current value of SRes. in our table is actually a typo. It should read $71.5\\%$ and not $81.4\\%$, which can be found in the row labeled "Alex – lin". All other values are correct.
> > >
> > > With this fix, the largest difference between LASI (ours) and fine-tuned LPIPS is now $3.6\\%$ in the "Trad." category.
> > >
> > > Despite being small, this difference is easily explained by the following quote from Table 5 of [1] (which applies to the first 2 columns in our table): "*LPIPS metrics are trained on the same traditional and CNN-based distortions, and as such have an advantage relative to other methods when testing on those same distortion types, even on unseen test images".
> > >
> > > [1] https://arxiv.org/abs/1801.03924

---

> > > ### Author Response · Authors · 2023-11-21
> > >
> > > We are sorry for the delay.
> > >
> > > A revision has been posted that includes the ablations, fixed table, and discussion regarding compute.
> > >
> > > Thank you once again for the attention to detail.

---

> ### Comment · Reviewer_8dYU · 2023-11-23
>
> Thanks for the efforts and clarifications that have addressed my concerns. Thus, I have updated the score accordingly.

---

### Official Review · Reviewer_WqhQ · 2023-11-08

**Soundness:** 3 good
**Presentation:** 3 good
**Contribution:** 2 fair
**Rating:** 6
**Confidence:** 3

**Summary:**

The paper focuses on generating perceptual embedding of an image, without using any training data. The paper also introduces and proposes a distance metric called LASI. The author goes on to compare the proposed method's performance with existing methods such as LPIPS and others.

**Strengths:**

- The paper looks to solve an important problem in the domain of computer vision which is to qualify the quality of embedding generated which matches with its perceptual quality, without using any training dataset.
- Conducts evaluation on the BAPPS dataset with other metrics such as LPIPS, PIM, and MS-SSIM.
- Achieves comparative and better results in some cases compared to current state-of-art methods.
- A good amount of side experiment details are shown to better verify the claims presented in the paper.
-The paper for the most part of it well organized without any obvious typos and a writing structure that is easy to follow.

**Weaknesses:**

- There is minimal discussion about the failure cases using the proposed method. Would be great to have some qualitative results and the probable reason we are seeing the results as we see it.
- Authors fails to discuss adequately why in some cases other metrics (such as PIM) excel compared to the LASI metric.

**Questions:**

1. In the results presented in Table:1, why does MS-SSIM outdo the author's proposed method for the BAPPS-JND task.. whereas it outperforms it for the BAPPS-2AFC task?
2. As mentioned in the weakness section, it is imperative that authors present more details qualitative and quantitative about the failure cases seen using the LASI method proposed in the paper.
3. In Section 5.3: "....Results indicate LASI can find failure points for LPIPS and vice-versa......" Can authors elaborate on this point ?

---

> ### Author Response · Authors · 2023-11-18
>
> We thank the reviewer for the questions regarding the surprising performance of LASI (ours) over neural network baselines. Indeed the performance of LASI (ours) is extremely close to  that of LPIPS and PIM, which is the central point of this paper.
>
> We hope our rebuttal addresses your concerns and you will consider increasing your score.
>
> ---
>
> > There is minimal discussion about the failure cases using the proposed method. Would be great to have some qualitative results and the probable reason we are seeing the results as we see it.
>
> > Authors fails to discuss adequately why in some cases other metrics (such as PIM) excel compared to the LASI metric.
>
> > As mentioned in the weakness section, it is imperative that authors present more details qualitative and quantitative about the failure cases seen using the LASI method proposed in the paper.
>
> The difference in performance between LASI (ours) and PIM is insignificant (less than $2.3\\%$ averaged across categories) leaving very little signal to perform a detailed analysis, which is the main point of the paper.
>
> A qualitative analysis on the examples of the dataset yields no immediately obvious pattern that would explain this small difference in performance.
>
> The comparison in Table 1 favors alternative metrics by design.
> LASI (ours) is held fixed while the best performing models for other metrics are chosen for each category.
> For example, for LPIPS we picked the best performing embedding between VGG, SqueezeNet, and AlexNet, for each category, while for PIM we chose between PIM-1 and PIM-5 (whichever scored higher).
> This cherry picking explains the gap: if we tune LASI (ours) by varying $\omega$, then the gap can be removed.
>
> This is exactly the main message of the paper: we can achieve perceptual performance virtually indistinguishable from that of neural network methods while using no training data or deep features, even when neural network methods are allowed to be fine-tuned for each category.
>
> ---
>
> > In the results presented in Table:1, why does MS-SSIM outdo the author's proposed method for the BAPPS-JND task.. whereas it outperforms it for the BAPPS-2AFC task?
>
> MS-SSIM does not outdo our method on the JND task.
> This is only true in one category of the JND task and by less than $1\\%$, while in the other category our method is superior by $54.4\\%$.
>
> In all other categories, for both 2AFC and JND, LASI (ours) outperforms MS-SSIM.
>
> This is highlighted in Table 1, in the last row labeled "Improv. over MS-SSIM", where only the last column is negative ($-0.9\\%$) and all others are positive.
>
> ---
>
> > In Section 5.3: "....Results indicate LASI can find failure points for LPIPS and vice-versa......" Can authors elaborate on this point ?
>
> Images in the box labeled “LPIPS loss fixed” have the same LPIPS distance to the reference.
> This means LPIPS says all images have equal perceptual quality relative to the reference.
> However, clearly the bottom row has better perceptual quality than the top row.
> This is known as a “failure mode” in the MAD literature.
>
> Conceptually, this is analogous to finding adversarial examples for a classifier – all known perceptual metrics have failure modes. In conducting this experiment, we were hoping to find qualitative explanations for the unexpected performance of LASI, but the results have so far been inconclusive.

---

> > ### Comment · Reviewer_WqhQ · 2023-11-21
> >
> > >>>> Authors fails to discuss adequately why in some cases other metrics (such as PIM) excel compared to the LASI metric.
> >
> > '''.....This cherry-picking explains the gap: if we tune LASI (ours) by varying ω, then the gap can be removed...'''
> > 1. Are we sure about this, If yes can authors present results for the same and show their method outperforming PIM-based evaluation?
> > 2. What are the probable disadvantages of choosing (a much refined ) ω, since the authors have refrained from choosing one till this point?
> > 3. Do authors use the same ω for a given task while comparing with other counterparts?
> >
> >
> > >>> In the results presented in Table:1, why does MS-SSIM outdo the author's proposed method for the BAPPS-JND task.. whereas it outperforms it for the BAPPS-2AFC task?
> > The results presented and as explained by the author more-or-so require an explanation of why is trend reversed across the same task, across Traditional and CNN, where you see an improvement of about 54% but a degradation of 1% ? How would the author  explain this ?

---

> > > ### Author Response · Authors · 2023-11-22
> > >
> > > > 1. Are we sure about this, If yes can authors present results for the same and show their method outperforming PIM-based evaluation?
> > >
> > >
> > > Yes. We performed an ablation on $\omega$ for BAPPS-2AFC to show the gap between PIM and LASI can be brought down to less than $0.5\\%$ on average, with LASI outperforming PIM in specific categories despite requiring significantly less resources.
> > >
> > >
> > > |      |  Trad. |   CNN  |   SRes.   | Deblur. | Color. |  Interp.  |
> > > |-----:|:------:|:------:|:---------:|:-------:|:------:|:---------:|
> > > |  PIM |  75.7  |  82.7  |    70.3   |   61.6  |  63.3  |    62.6   |
> > > | LASI |  74.6  |  82.5  |    70.4   |   61.1  |   63   |    62.9   |
> > > |  Gap | -1.45% | -0.24% | **0.14%** |  -0.81% | -0.47% | **0.48%** |
> > >
> > >
> > > We have run out of GPU resources to run the BAPPS-JND experiments, but will include them in the appendix of the paper in the camera-ready.
> > >
> > >
> > > ---
> > >
> > >
> > > > 2. What are the probable disadvantages of choosing (a much refined ) ω, since the authors have refrained from choosing one till this point?
> > >
> > >
> > > Smaller values of $\omega$ will bias the solution of Equation (1) to favor reconstruction of closer points when building the embeddings of each pixel. Ideally this should be tuned to the spatial frequency of the image. For example, images that vary quickly benefit from a smaller $\omega$. However, we found the default value of $0.8$ worked well across all BAPPS categories. The computational and memory requirements to run LASI are independent of the choice of $\omega$. We will include this discussion in the main text.
> > >
> > >
> > > ---
> > >
> > >
> > > > 3. Do authors use the same ω for a given task while comparing with other counterparts?
> > >
> > >
> > > Yes, $\omega=0.8$ was held fixed across all experiments. This is the value used in the original lossless compression algorithm that inspired our method.
> > >
> > >
> > > ---
> > >
> > > > The results presented and as explained by the author more-or-so require an explanation of why is trend reversed across the same task, across Traditional and CNN, where you see an improvement of about 54% but a degradation of 1% ? How would the author explain this ?
> > >
> > > The low performance of MS-SSIM on “Traditional” distortions, for both 2AFC and JND, is not unexpected. Previous literature [1, 2] has shown that MS-SSIM does not perform well on “geometric” transformations (e.g., warping, and shifting) which are heavily present in this category (see Table 2 of [3] for the complete list of distortions). Meanwhile, CNN distortions (also described in Table 2 of [3]) are more easily detectable as it modifies the structure of the image, resulting in a much smaller difference in performance between all methods. We will include this discussion in the text.
> > >
> > > [1] https://arxiv.org/abs/2006.06752
> > >
> > > [2] https://arxiv.org/abs/2005.01338
> > >
> > > [3] https://arxiv.org/abs/1801.03924
> > >
> > > ---
> > >
> > >
> > > > Authors fails to discuss adequately why in some cases other metrics (such as PIM) excel compared to the LASI metric.
> > >
> > > PIM requires training data, deep features, is slower at inference time compared to our method (see Figure 3), and outperforms LASI by only $2.3%$ on average. As we have shown in the ablations above, tuning the hyperparameter $\omega$ brings this gap down to less than $0.5\\%$ and makes LASI outperform PIM in some categories. We consider the gap to be too small to be significant.
> > >
> > > LPIPS is trained on the same distortion types present in “Trad.” and “CNN” and therefore has an unfair advantage relative to other metrics in these categories, as mentioned in Zhang et al., 2018.
> > >
> > > We have added this discussion to the main text in Table 1.

---

> > > > ### Comment · Reviewer_WqhQ · 2023-11-22
> > > >
> > > > I am fairly satisfied with the author's response. I would like to maintain my rating of marginally above accept.

---

### Author Response · Authors · 2023-11-18
**General response**

We would like to thank the reviewers for the thoughtful questions and acknowledgement that the great performance of our method (LASI) is indeed a surprising result.

We summarize the main questions here together with a short response for the benefit of the AC.

---

> How does the compute time of LASI (ours) compare to DNN methods?

LASI is overall faster than DNN at inference time, for all experiments considered, as shown in the plot below.

[Please click for LASI and PIM comparison plot](https://postimg.cc/ygnnh7Vg)

---

> It seems $N$ needs to be tuned for each image resolution, which can be critical, since images can come in at various sizes.

LASI does not need to be tuned for each resolution, as the performance on perceptual tasks increases with $N$. Instead, $N$ should be made as large as possible while respecting the compute and memory budget of the application.

---

> Why does MS-SSIM outperform our method on the JND tasks?

MS-SSIM does not outdo our method on the JND task.
This is true only on a single category of JND where the difference is insignificant ($-0.9\\%$).
On all other tasks our method outperforms MS-SSIM - and by a wide margin (more than $50\\%$) in some cases.

---

### Meta-Review · Area_Chair_Ut6H · 2023-12-10

**Metareview:**

The authors address the task of image quality assessment and propose Linear Autoregressive Similarity Index (LASI) which uses weighted least squares to obtain perceptual embeddings and is inspired by a lossless compression algorithm. The data-free LASI is shown to provide comparable performance with supervised (eg LPIPS) and unsupervised methods (PIM).

The idea is novel, the paper reads well, and the experimental results support the claims. On the downside, the authors could have invested more into failure cases and reporting on more datasets.

The authors provided responses to all the reviewers' concerns and the reviewers are generally unanimously leaning towards acceptance of the paper (6,6,6,6).

The meta-reviewer after carefully reading the reviews, the discussions, and the paper, agrees with the reviewers and recommends acceptance.

**Justification For Why Not Higher Score:**

While the idea and the performance of the method are strengths, the advances at the machine learning or theoretical level are limited and the interest for the addressed topic is rather limited within the ICLR community.

**Justification For Why Not Lower Score:**

Four reviewers and the meta-reviewer agree that the paper has merits, and the contributions are sufficient and of interest. There is no significant flaw to impede publication.

---

### Decision · Program_Chairs · 2024-01-16

Accept (poster)